# Non-linear association between aspartate aminotransferase to alanine aminotransferase ratio and mortality in critically ill older patients: A retrospective cohort study

**Hua Yang[1,2]☯, Limin Meng[1,2]☯, Shuanli Xin[2], Chao Chang[2], Xiufeng Zhao[2], Bingyan Guo[1] ***

1 Department of Cardiology, The Second Hospital of Hebei Medical University, Shijiazhuang, Hebei, China,
2 Department of Cardiology, Handan First Hospital, Handan, Hebei, China

☯ These authors contributed equally to this work.
* gby202201@163.com

## Abstract

### Background

The aspartate aminotransferase to alanine aminotransferase (AST/ALT) ratio has been shown to be associated with poor clinical outcomes across various patient groups. However, little is unclear about the association between the two in critically ill older patients. Therefore, we aim to investigate the association of the AST/ALT ratio with hospital mortality in this special population.

### Methods

In this retrospective cohort study, we extracted elderly patients (age ≥ 65 years) from the Medical Information Mart for Intensive Care IV (MIMIC-IV) database. The primary outcome was in-hospital mortality. The association between the AST/ALT ratio and hospital mortality was studied using univariable and multivariable Cox regression analysis, as well as restricted cubic splines (RCS). Survival analysis was performed using the Kaplan-Meier (KM) method according to the AST/ALT ratio.

### Results

Among the 13,358 eligible patients, the mean age was 77.6 years, 7,077 patients (52.9%) were male, and 2,511 patients (18.8%) died in hospital. The AST/ALT ratio was found to be independently associated with in-hospital mortality (HR = 1.05, 95% CI: 1.01–1.09, $P$ = 0.022) after adjusting for potential confounders. Furthermore, a non-linear relationship and saturation effect were observed between them, with the inflection point being 1.80. When the AST/ALT ratio was less than 1.80, we found that every 1 unit increase in the AST/ALT ratio resulted in a 39% increased risk of in-hospital mortality (HR = 1.39, 95% CI: 1.18–1.64,

**Data Availability Statement:** All datasets supporting the conclusions of the present study

are obtained from the MIMIC-IV database (web site: https://physionet.org/content/mimiciv/).

**Funding:** The authors received no specific funding for this work.

**Competing interests:** The authors have declared that no competing interests exist.

*P* < 0.001). However, when the AST/ALT ratio was greater than 1.80, the association became saturated (HR = 1.01, 95% CI: 0.96–1.07, *P* = 0.609). Sensitivity and subgroup analyses showed the results were robust.

## Conclusion

In critically ill older patients, the association between the AST/ALT ratio and in-hospital mortality was non-linear and showed a saturation effect. An elevated AST/ALT ratio was significantly associated with increased in-hospital mortality when the AST/ALT ratio was less than 1.80.

## Introduction

Advances in modern medicine mean many people can expect to live far longer than in previous generations. As the population ages, there will be more than 700 million people over the age of 65 in 2020, with the number expected to double by 2050 [1]. Global aging trends will have major impacts on health and social care systems, so aging populations are a hot topic. Furthermore, as science in medicine and disease management has advanced, there has been a dramatically increase in the number of older critically ill patients being admitted to the intensive care unit (ICU) [2, 3]. Critically ill older patients make up a significant proportion of the ICU. According to a global multi-center survey conducted in 2017, the mean age of ICU patients is 60 years old [4]. Additionally, in many countries, the median age of the general ICU population exceeds 65 [5]. Older individuals often experience decreased immune functioning [6], a decline in organ function, and the exacerbation of various comorbidities. This process can be further influenced by a range of social, psychological, and economic factors [7]. Consequently, older patients face a greater probability of adverse outcomes, such as mortality [8]. Additionally, healthcare costs for elderly ICU patients are significantly higher than for their younger counterparts, which can place a considerable burden on families and society [9]. Therefore, critically ill older patients demand extra attention and care from physicians. In recent years, factors impacting the outcome of critically ill older patients have become an important element of research.

Aspartate aminotransferase (AST) is present in both the cytoplasm and mitochondria of tissue cells, such as the liver, skeletal muscle, heart, brain, and even the kidney, whereas alanine transaminase (ALT) is predominantly located in the cytoplasm of liver tissue. It is thought that ALT primarily reflects liver-specific malfunction, whereas AST may indicate mitochondrial dysfunction resulting from oxidative stress in other tissues to a certain degree [10]. Increases in the AST/ALT ratio are not only linked to liver fibrosis severity and liver function impairment [11], but might also play a role in more systemic derangements such as increased oxidative stress, inflammatory response, and ischemic-reperfusion injury [12]. Previous research has revealed that the AST/ALT ratio could function as a substitute indicator for ischemic end-organ damage during the acute phase and be linked to malnutrition, frailty, and mortality among elderly individuals [13, 14]. An increased AST/ALT ratio has been linked to worse outcomes in patients with a range of health conditions, including heart failure [15, 16], acute myocardial infarction [17, 18], cardiac arrest [19], sepsis [20], hypertension [21], type 2 diabetes [10], and COVID-19 [22, 23], among others.

However, to the best of our knowledge, there is currently no research exploring the clinical association between the AST/ALT ratio and mortality among critically ill older patients.

Therefore, the objective of this study is to investigate the association between the AST/ALT ratio and hospital mortality within this group.

## Materials and methods

### Data source

Data was extracted from the Medical Information Mart for Intensive Care IV (MIMIC-IV version 2.0), a publicly accessible database containing information on over 52,000 individuals admitted to the Beth Israel Deaconess Medical Center between 2008 and 2019 [24]. To access this database, we successfully completed the Protecting Human Research Participants training course (certificate number: 48851221). All patient-related information in the database is anonymous. The study was approved and granted a waiver of informed consent by the institutional review boards of the Massachusetts Institute of Technology (MIT) and Beth Israel Deaconess Medical Center (BIDMC). The study adhered to the Declaration of Helsinki guidelines and complied with the Strengthening the Reporting of Observational Studies in Epidemiology (STROBE) reporting guidelines.

### Study population

This was a single-center retrospective cohort study conducted on older adult patients (aged 65 years and above) admitted to the ICU. We excluded patients whose AST or ALT records were missing on the first day of ICU admission. Moreover, we only analyzed data from their initial ICU stay for patients with multiple admissions to the ICU. In total, 13,358 eligible patients were included, and they were categorized into three groups according to their AST/ALT ratio tertiles on their first ICU stay. Fig 1 depicts a patient screening flow chart.

### Study variables

Data was extracted using Structured Query Language (SQL) and PostgreSQL (version 14.0). The data included demographic information, vital signs, biochemistry and laboratory indicators, disease severity upon admission, and comorbidity. The AST/ALT ratio was determined by dividing the respective values of AST and ALT (both in IU/L). For variables recorded more than once in the initial 24 hours, the mean value was utilized. In addition, variables with more than 10% missing values were excluded. Comorbidities were determined using documented International Classification of Diseases (ICD) 9 and 10 diagnostic codes. The list of comorbidities included hypertension, diabetes mellitus, atrial fibrillation or flutter, myocardial infarction, acute kidney injury, liver disease, peripheral vascular disease, congestive heart failure, cerebrovascular disease, chronic pulmonary disease, cardiac arrest, and cardiogenic shock. Sepsis was diagnosed using the "Third International Consensus Definition for Sepsis and Septic Shock" (sepsis-3).

### Primary outcome

The primary outcome of this study was all-cause mortality during hospitalization, which was defined as death during hospitalization. ICU and general ward inpatient deaths were among the hospital deaths. There were no patients who were lost to follow-up.

### Statistical analysis

Continuous variables were reported as means ± standard deviations, while categorical variables as n (%). When continuous variables did not satisfy a normal distribution, medians and

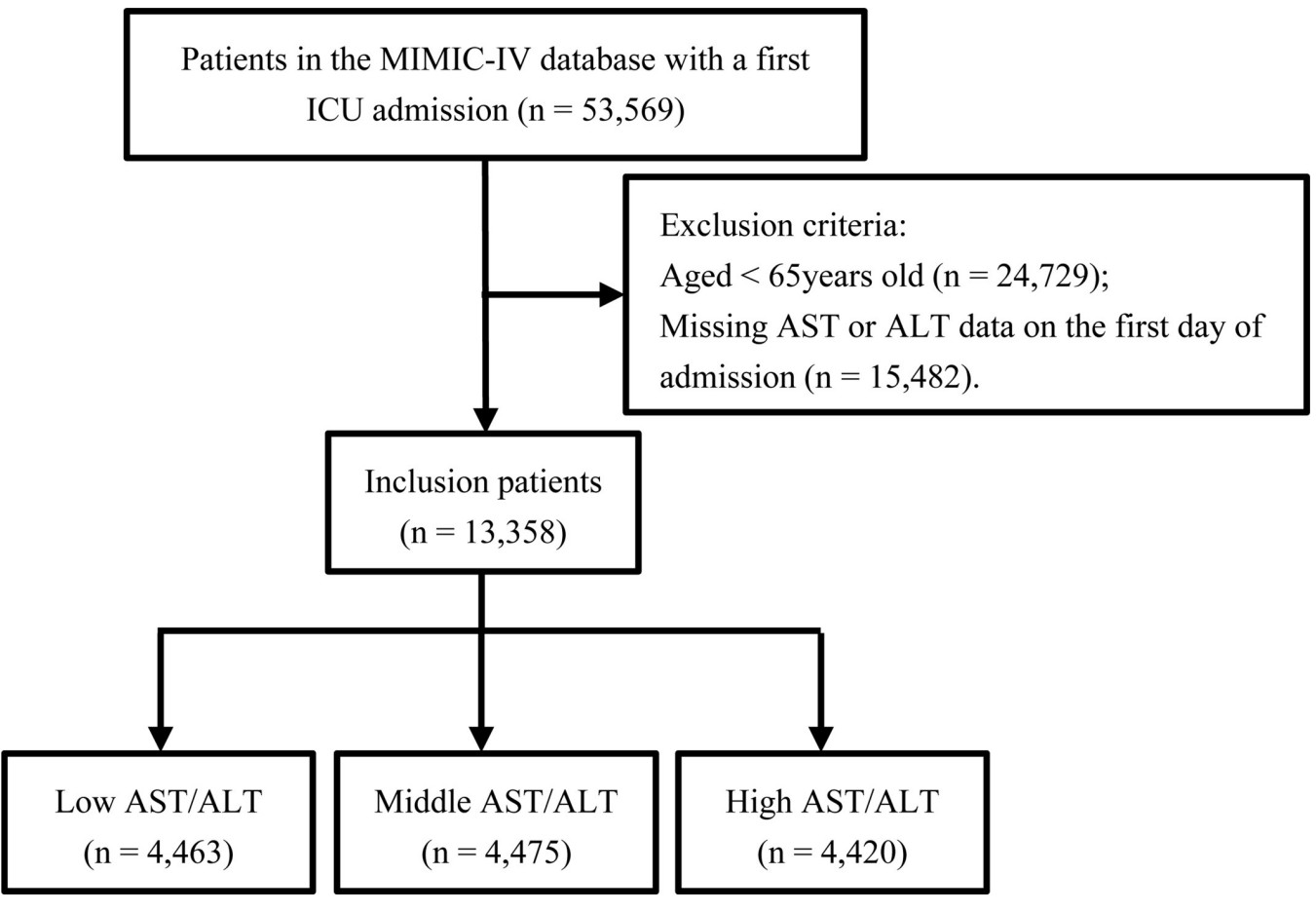

**Fig 1. Flowchart of study selection.**

interquartile ranges (IQR) were utilized instead. We used One-way ANOVA, Chi-square test, and Kruskal-Wallis H test to compare differences in baseline characteristics among groups.

We compared survival outcomes among AST/ALT ratio groups using Kaplan-Meier survival analysis and the log-rank test. Meanwhile, we examined the association between the AST/ALT ratio and mortality by utilizing Cox proportional hazards regression models, with the results expressed as hazard ratios (HR) and 95% confidence intervals (CI). Our selection of confounders was based on their associations with mortality or an effect estimate change exceeding 10%. We reported three models: Model 1 (unadjusted), Model 2 (age and gender adjustments), and Model 3 (age, gender, smoking, alcohol, weight, systolic blood pressure, respiratory rate, liver disease, cerebrovascular disease, cardiac arrest, cardiogenic shock, acute kidney injury, sequential organ failure assessment, systemic inflammatory response syndrome, hemoglobin, platelets, anion gap, blood urea nitrogen, and potassium adjustments). The variance inflation factor (VIF) was used to determine parameters collinearity, with a VIF greater than 5 indicating the presence of multicollinearity. To check the proportional hazards assumption, we utilized the "cox.zph" function from the R (survival) package. There were missing covariate values ranging from 0.04% for blood urea nitrogen to 1.13% for bicarbonate, which we indicated using dummy variables [25]. In addition, we conducted sensitivity analyses by introducing missing data as separate dummy variables into the original dataset. Moreover, we used a restricted cubic spline (RCS) regression model to evaluate the non-linear relationship

between the AST/ALT ratio and in-hospital mortality, and a two-piecewise linear regression model to determine the inflection point. To examine the one-line linear regression model against the two-piecewise linear model, we conducted a log likelihood ratio test. Additionally, we performed subgroup analyses using stratified Cox proportional hazards regression models, and examined the modification and interaction of subgroups using a likelihood ratio test.

R software 3.6.3 was used to conduct all analyses. Statistical significance was defined as a two-sided $P$ value of 0.05.

## Results

### Baseline characteristics

Among the 13,358 patients in our study, 52.9% (7,077) were male, and their average age was 77.6 years. The median levels of AST and ALT were 36.0 IU/L (23.0–79.0) and 24.0 IU/L (15.0–52.0), respectively, along with a median AST/ALT ratio of 1.5 (1.1–2.1). The average follow-up time for our study was 12.9 days. Throughout the study, 18.8% (2,511) of the patients experienced all-cause mortality, with 12.6% (1683) of them occurring during the ICU stay.

The patients were categorized into three groups according to their AST/ALT ratio tertiles: low ($\leq$ 1.24, n = 4,463), middle (1.25–1.83, n = 4,475), and high ($>$ 1.83, n = 4,420). Table 1 describes the baseline characteristics. Patients with a higher AST/ALT ratio were lower in weight, more likely to drink alcohol, had higher severity of illness scores on admission, lower systolic blood pressure, and a higher prevalence of various medical conditions, including myocardial infarction, liver disease, sepsis, acute kidney injury, cardiac arrest, congestive heart failure, peripheral vascular disease, and cardiogenic shock. They also had higher levels of AST, white blood cells, creatinine, anion gap, and potassium, and lower levels of ALT, hematocrit, hemoglobin, platelets, and sodium. Ultimately, those with high AST/ALT ratios had a higher in-hospital and ICU mortality rate and a longer hospital and ICU stay.

### Survival analysis

The Kaplan-Meier survival curves that illustrate the relationship between AST/ALT ratio tertiles and in-hospital mortality are shown in Fig 2. The mortality rate differed statistically across groups (low AST/ALT ratio: 13.6% vs. middle AST/ALT ratio: 18.3% vs. high AST/ALT ratio: 24.6%, $P <$ 0.001). Notably, patients with elevated AST/ALT ratios exhibited a significantly increased risk of in-hospital mortality.

### Association of the AST/ALT ratio with mortality

Cox proportional hazards regression analysis demonstrated a significant association between the AST/ALT ratio and hospital mortality in the unadjusted model (HR = 1.18, 95% CI: 1.14–1.22, $P <$ 0.001). The association did not change significantly after excluding 272 patients (2.0%) with missing data and adjusting for potential confounders (Model 3); the association of the AST/ALT ratio with hospital mortality remained significant (HR = 1.05, 95% CI:1.01–1.09, $P =$ 0.022) (Table 2). Furthermore, we also performed sensitivity analysis using dummy variables to indicate missing data and obtained similar results in Model 3 (HR = 1.04, 95% CI:1.00–1.07, $P =$ 0.040) (S1 Table).

To conduct sensitivity analysis, we categorized the AST/ALT ratio and used low AST/ALT as a reference group for comparison. In Model 3, patients with middle and high AST/ALT ratios had a 13% and 17% increased risk of mortality, respectively (HR = 1.13, 95% CI: 1.01–1.25, $P =$ 0.031; HR = 1.17, 95% CI: 1.05–1.30, $P =$ 0.004; Table 2). Furthermore, analyses using dummy variables to indicate missing data yielded similar results.

**Table 1. Baseline characteristics of participants.**

| Characteristics | Total (n = 13,358) | AST/ALT ratio | | | P value |
|---|---|---|---|---|---|
| | | ≤ 1.24 (n = 4,463) | 1.25–1.83 (n = 4,475) | > 1.83 (n = 4,420) | |
| **Demographic** | | | | | |
| Age, years, | 77.6 ± 8.2 | 76.6 ± 7.9 | 78.3 ± 8.3 | 78.0 ± 8.3 | < 0.001 |
| Male, n (%) | 7,077 (52.9) | 2,533 (56.8) | 2,284 (51.0) | 2,260 (51.1) | < 0.001 |
| Weight, kg, | 77.7 ± 21.5 | 80.0 ± 22.7 | 77.0 ± 20.2 | 76.2 ± 21.3 | < 0.001 |
| Smoking, n (%) | 5,471 (41.0) | 1,915 (42.9) | 1,755 (39.2) | 1,801 (40.7) | 0.002 |
| Alcoholic, n (%) | 867 (6.5) | 219 (4.9) | 295 (6.6) | 353 (8.0) | <0.001 |
| **ICU admission** | | | | | |
| SOFA score | 5.8 ± 3.9 | 5.1 ± 3.5 | 5.6 ± 3.8 | 6.6 ± 4.1 | < 0.001 |
| SIRS score | 2.5 ± 1.0 | 2.5 ± 1.0 | 2.5 ± 1.0 | 2.6 ± 1.0 | < 0.001 |
| SAPSII | 42.3 ± 13.6 | 40.3 ± 12.7 | 41.9 ± 13.4 | 44.9 ± 14.4 | < 0.001 |
| Charlson comorbidity index | 7.2 ± 2.4 | 7.0 ± 2.4 | 7.1 ± 2.4 | 7.4 ± 2.5 | < 0.001 |
| **Vital signs** | | | | | |
| HR, bpm | 83.6 ± 16.4 | 83.1 ± 16.7 | 83.1 ± 16.2 | 84.7 ± 16.1 | <0.001 |
| SBP, mmHg | 118.9 ± 17.7 | 120.9 ± 17.6 | 119.6 ± 17.6 | 116.3 ± 17.5 | <0.001 |
| DBP, mmHg | 61.6 ± 10.9 | 62.9 ± 10.7 | 61.5 ± 10.8 | 60.3 ± 10.8 | <0.001 |
| Respiratory rate, bpm | 19.8 ± 3.8 | 19.7 ± 3.7 | 19.7 ± 3.7 | 19.9 ± 3.9 | 0.014 |
| **Comorbidities, n (%)** | | | | | |
| Myocardial infarct | 3,038 (22.7) | 869 (19.5) | 915 (20.4) | 1,254 (28.4) | < 0.001 |
| Congestive heart failure | 4,930 (36.9) | 1,579 (35.4) | 1,618 (36.2) | 1,733 (39.2) | < 0.001 |
| Peripheral vascular disease | 1,884 (14.1) | 580 (13) | 612 (13.7) | 692 (15.7) | < 0.001 |
| Cerebrovascular disease | 2,648 (19.8) | 859 (19.2) | 963 (21.5) | 826 (18.7) | 0.002 |
| Chronic pulmonary disease | 3,648 (27.3) | 1,294 (29.0) | 1,182 (26.4) | 1,172 (26.5) | 0.008 |
| Liver disease | 1,533 (11.5) | 372 (8.3) | 490 (10.9) | 671 (15.2) | < 0.001 |
| Sepsis | 7,595 (56.9) | 2,389 (53.5) | 2,496 (55.8) | 2,710 (61.3) | < 0.001 |
| Acute kidney injury | 8,612 (64.5) | 2,590 (58) | 2,848 (63.7) | 3,174 (71.8) | < 0.001 |
| Atrial fibrillation or flutter | 5,477 (41.0) | 1,791 (40.1) | 1,820 (40.7) | 1,866 (42.2) | 0.116 |
| Diabetes mellitus | 4,484 (33.6) | 1,611 (36.1) | 1,479 (33.1) | 1,394 (31.5) | < 0.001 |
| Hypertension | 6,180 (46.3) | 2,184 (48.9) | 2,106 (47.1) | 1,890 (42.8) | < 0.001 |
| Cardiac arrest | 713 (5.3) | 201 (4.5) | 238 (5.3) | 274 (6.2) | 0.002 |
| Cardiogenic shock | 924 (6.9) | 221 (5.0) | 274 (6.1) | 429 (9.7) | < 0.001 |
| **Laboratory tests** | | | | | |
| AST, IU/L | 36.0 (23.0, 79.0) | 31.0 (20.0,65.0) | 31.0 (22.0, 61.0) | 50.0 (30.0,112.0) | < 0.001 |
| ALT, IU/L | 24.0 (15.0, 52.0) | 35.0 (21.0,80.0) | 21.0 (15.0, 41.0) | 19.0 (12.0, 39.0) | < 0.001 |
| AST/ALT ratio | 1.5 (1.1, 2.1) | 1.0 (0.8, 1.1) | 1.5 (1.4, 1.6) | 2.4 (2.1, 3.1) | < 0.001 |
| WBC, $10^9$/L | 12.4 (8.8, 17.4) | 11.9 (8.7, 16.6) | 12.1 (8.8, 17.2) | 12.9 (9.1, 18.6) | < 0.001 |
| Hematocrit, % | 35.0 ± 6.3 | 35.3 ± 6.4 | 35.0 ± 6.1 | 34.6 ± 6.3 | < 0.001 |
| Hemoglobin, g/dL | 11.4 ± 2.1 | 11.6 ± 2.2 | 11.4 ± 2.1 | 11.3 ± 2.1 | < 0.001 |
| Platelets, $10^9$/L | 208.0 (154.0,278.0) | 212.0 (159.0,278.0) | 211.0 (156.0,282.0) | 202.0 (148.0,272.0) | < 0.001 |
| Anion gap, mEq/L | 17.4 ± 5.0 | 16.7 ± 4.5 | 17.3 ± 4.8 | 18.2 ± 5.6 | < 0.001 |
| Bicarbonate, mEq/L | 24.5 ± 4.1 | 24.5 ± 4.0 | 24.5 ± 4.1 | 24.4 ± 4.1 | 0.747 |
| Calcium, mg/dL | 8.6 ± 0.9 | 8.7 ± 0.8 | 8.7 ± 0.8 | 8.6 ± 1.1 | 0.276 |
| Chloride, mEq/L | 105.6 ± 6.6 | 105.6 ± 6.7 | 105.7 ± 6.5 | 105.4 ± 6.6 | 0.061 |
| Sodium, mEq/L | 140.0 ± 5.4 | 140.2 ± 5.6 | 140.1 ± 5.3 | 139.8 ± 5.4 | 0.002 |
| Potassium, mEq/L | 4.6 ± 0.9 | 4.5 ± 0.8 | 4.6 ± 0.8 | 4.8 ± 1.0 | < 0.001 |
| Blood urea nitrogen, mg/dL | 27.0 (18.0, 43.0) | 27.0 (18.0, 43.0) | 26.0 (18.0, 42.0) | 28.0 (19.0, 44.0) | 0.006 |
| Creatinine, mg/dL | 1.2 (0.9, 1.9) | 1.2 (0.9, 1.7) | 1.2 (0.9, 1.8) | 1.3 (0.9, 2.1) | < 0.001 |

*(Continued)*

**Table 1.** (Continued)

| Characteristics | Total (n = 13,358) | AST/ALT ratio | | | P value |
|---|---|---|---|---|---|
| | | ≤ 1.24 (n = 4,463) | 1.25–1.83 (n = 4,475) | > 1.83 (n = 4,420) | |
| Glucose, mg/dL | 150.0 (120.0,202.0) | 152.0 (120.0,207.0) | 150.0 (119.0,201.0) | 150.0 (119.0,199.0) | 0.046 |
| **Outcomes** | | | | | |
| In-hospital death, n (%) | 2,511 (18.8) | 605 (13.6) | 818 (18.3) | 1,088 (24.6) | < 0.001 |
| ICU death, n (%) | 1,683 (12.6) | 380 (8.5) | 532 (11.9) | 771 (17.4) | < 0.001 |
| LOS hospital, days | 7.0 (4.0, 13.0) | 7.0 (4.0, 12.0) | 7.0 (4.0, 12.0) | 8.0 (4.0, 14.0) | < 0.001 |
| LOS ICU, days | 2.2 (1.2, 4.2) | 2.0 (1.1, 3.9) | 2.1 (1.2, 4.0) | 2.4 (1.4, 4.8) | < 0.001 |

**Notes:** Data are presented as the mean ± SD, median (IQR), and n (%).

**Abbreviations**: SOFA, sequential organ failure assessment; SIRS, systemic inflammatory response syndrome; SAPSII, simplified acute physiological score II; HR, heart rate; bmp, beats per minute; SBP, systolic blood pressure; DBP, diastolic blood pressure; AST, aspartate aminotransferase; ALT, alanine aminotransferase; WBC, white blood cell; ICU, intensive care unit; LOS, length of stay.

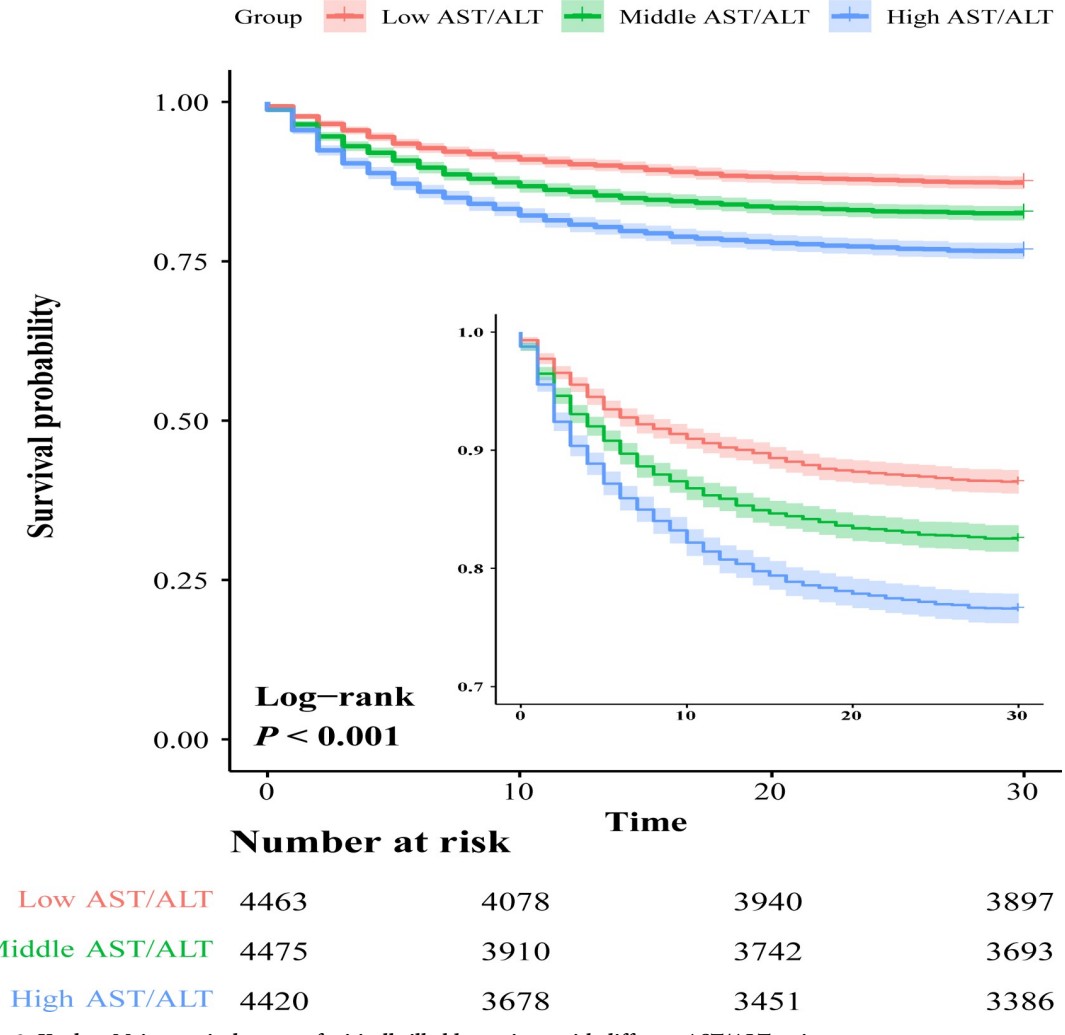

**Fig 2. Kaplan–Meier survival curves of critically ill older patients with different AST/ALT ratios.**

**Table 2. Association between AST/ALT ratio and the risk of mortality in different models.**

| Variable | Model 1 | | Model 2 | | Model 3 | |
|---|---|---|---|---|---|---|
| | HR (95% CI) | *P* value | HR (95% CI) | *P* value | HR (95% CI) | *P* value |
| AST/ALT ratio | 1.18 (1.14~1.22) | < 0.001 | 1.17 (1.13~1.21) | < 0.001 | 1.05 (1.01~1.09) | 0.022 |
| AST/ALT ratio group | | | | | | |
| ≤ 1.24 | Reference | | Reference | | Reference | |
| 1.25–1.83 | 1.34 (1.21, 1.49) | < 0.001 | 1.28 (1.15, 1.42) | < 0.001 | 1.13 (1.01, 1.25) | 0.031 |
| > 1.83 | 1.66 (1.50, 1.83) | < 0.001 | 1.60 (1.45, 1.77) | < 0.001 | 1.17 (1.05, 1.30) | 0.004 |

**Notes:** Cox proportional hazards regression models were used to calculate hazard ratios (HR) with 95% confidence intervals (CI); Model 1 adjusted for: none; Model 2 adjusted for: age and gender; Model 3 adjusted for: age, gender, smoking, alcoholic, weight, systolic blood pressure, respiratory rate, liver disease, cerebrovascular disease, cardiac arrest, cardiogenic shock, acute kidney injury, sequential organ failure assessment, systemic inflammatory response syndrome, hemoglobin, platelets, anion gap, blood urea nitrogen, and potassium.

## The analysis of the non-linear association

In this study, we discovered a non-linear association between the AST/ALT ratio and the risk of hospital mortality after adjusting for potential confounders (*P* for non-linearity = 0.006, Fig 3). We used the two-piecewise linear regression model to determine the AST/ALT ratio inflection point, which was found to be 1.80 (log-likelihood ratio test: *P* < 0.001, Table 3). When the AST/ALT ratio was less than 1.80, every 1 unit increase in the AST/ALT ratio resulted in a 39% increased risk of in-hospital mortality (HR = 1.39, 95% CI: 1.18–1.64, *P* < 0.001), while on the right side, the relationship became saturated (HR = 1.01, 95% CI: 0.96–1.07, *P* = 0.609).

In addition, we divided the total population into two groups: one with normal values of liver function indicators (AST ≤ 40 U/L and ALT ≤ 40 U/L) and the other with abnormal liver function indicators (AST > 40 U/L or ALT > 40 U/L). We have evaluated the non-linear relationship and used a two-piecewise linear regression model to determine the inflection point. Similarly, a non-linear relationship and saturation effect were observed between them, respectively. The results were listed in S2 and S3 Tables and S1 and S2 Figs.

## Subgroup analysis

In addition, subgroup analyses were conducted for patients with AST/ALT ratio less than 1.80 to study the relationship between the AST/ALT ratio and in-hospital mortality (Fig 4). Stratifications were performed based on age and weight median, gender, status as a smoker or alcoholic, and the presence of liver disease. Except for the stratification component, each stratification was adjusted for all factors in Model 3. The findings of the subgroup analysis were quite consistent with the multivariable Cox regression analysis results. No significant interactions were observed.

## Discussion

Our retrospective cohort study revealed an independent association between the AST/ALT ratio and increased in-hospital mortality in critically ill older patients after adjustment for potential confounding factors. Furthermore, a non-linear association and saturation effect were observed between the two in further analysis. When the AST/ALT ratio was less than 1.80, there was an increase in the risk of in-hospital mortality with an increase in AST/ALT. However, on the right side of the inflection point, the association reached saturation. Additionally, subgroup and sensitivity analyses showed the results to be generally robust.

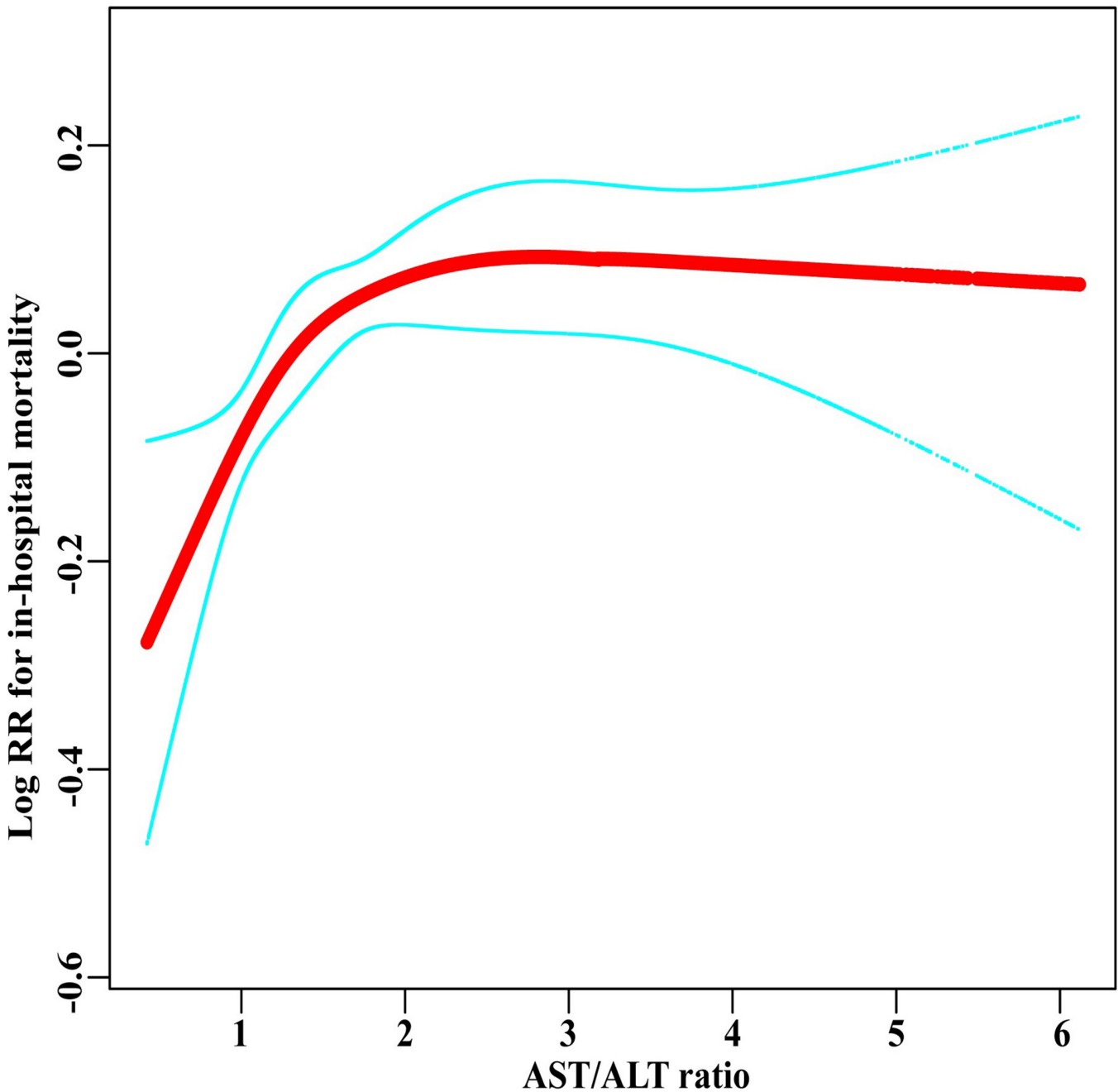

**Fig 3. The non-linear association between the ASL/ALT ratio and risk of mortality.** Adjusted for all factors in Model 3. The red lines represent the estimated values, and green lines represent their corresponding 95% confidence intervals.

The population admitted to the ICU has undergone significant changes in recent years. Because of the newly accessible sophisticated life support, an increasing number of older and more severely ill patients are receiving treatment in this environment. However, such patients are associated with elevated all-cause mortality rates [26]. In our study, the in-hospital mortality rate for critically ill older patients was found to be 18.8%, and this mortality was similar to other investigations reported. The cost of healthcare is also significantly higher for elderly ICU patients than for younger ICU patients [9]. Therefore, it is crucial to understand the factors

**Table 3. Threshold effect analysis of AST/ALT ratio on mortality using two-piecewise linear regression.**

|  | HR (95% CI) | P value |
|---|---|---|
| Fitting model by standard linear regression | 1.05 (1.01 ∼ 1.09) | 0.022 |
| Fitting model by two-piece wise linear regression |  |  |
| Inflection point of the AST/ALT ratio | 1.80 |  |
| ≤ 1.80 | 1.39 (1.18 ∼ 1.64) | < 0.001 |
| > 1.80 | 1.01 (0.96 ∼ 1.07) | 0.609 |
| P for log likelihood ratio test |  | < 0.001 |

**Notes:** We adjusted for: age, gender, smoking, alcoholic, weight, systolic blood pressure, respiratory rate, liver disease, cerebrovascular disease, cardiac arrest, cardiogenic shock, acute kidney injury, sequential organ failure assessment, systemic inflammatory response syndrome, hemoglobin, platelets, anion gap, blood urea nitrogen, and potassium.

**Abbreviations:** HR, hazard ratios; CI, confidence intervals.

that may influence mortality in critically ill older patients. It can help identify high-risk patients for targeted early intervention.

According to recent research, ALT is a more specific measure for the liver compared to AST. AST tends to increase or remain stable with age, whereas ALT generally decreases in elderly populations [27, 28]. Several studies have reported that low ALT levels in older individuals can be considered a marker of age-related frailty, disability, malnutrition, and sarcopenia, and are associated with higher all-cause mortality [14, 29]. This finding could be explained by age-related decline, which subsequently increases susceptibility to diseases. By contrast, AST is known to be an enzyme released from different tissues, reflecting anaerobic glycolysis. People with increased AST levels may have more severe tissue damage. A recent meta-analysis study with aggregate data found a comparatively moderate association between AST and all-cause mortality in general populations [30]. However, another study showed AST was positively associated with long-term mortality in this elderly population [31]. As a result, decreased ALT and increased AST levels could result in a high AST/ALT ratio. As observed in this study, there is an increasing pattern in median AST levels and a decreasing pattern in median ALT levels for patients whose AST/ALT ratios increase.

The AST/ALT ratio has been associated with oxidative stress and systemic inflammation [12]. Despite being initially suggested as a marker for liver function impairment severity, recent studies have found that the AST/ALT ratio is linked to mortality in a variety of patient subgroups, including myocardial infarction, cardiac arrest, heart failure, sepsis, hypertension, type 2 diabetes, and even COVID-19 patients [10, 15–23]. For example, in patients with acute myocardial infarction, an elevated AST/ALT ratio has been linked to both short- and long-term mortality risks [17, 18]. For acute and older heart failures, Maeda et al. [15, 16] demonstrated that the AST/ALT ratio may be associated with malnutrition, frailty, cachexia, and poor clinical outcomes. In 374 cardiac arrest patients, Lu et al. [19] discovered that a high AST/ALT ratio was found to be an independent predictor of ICU and in-hospital mortality. Schupp et al. [20] also reported a similar association of an elevated AST/ALT ratio with a higher risk of 30-day mortality in sepsis and septic shock patients. COVID, which is a disease associated with multiorgan involvement and ICU mortality in recent years. According to the former study, it showed that increasing AST and an initial AST to ALT ratio >1.5 are associated with mortality in hospitalized COVID-19 patients and may be an early warning [23]. Other studies have also demonstrated similar results in different patient groups [21, 22]. In agreement with these findings, we found a significant difference in the in-hospital mortality of

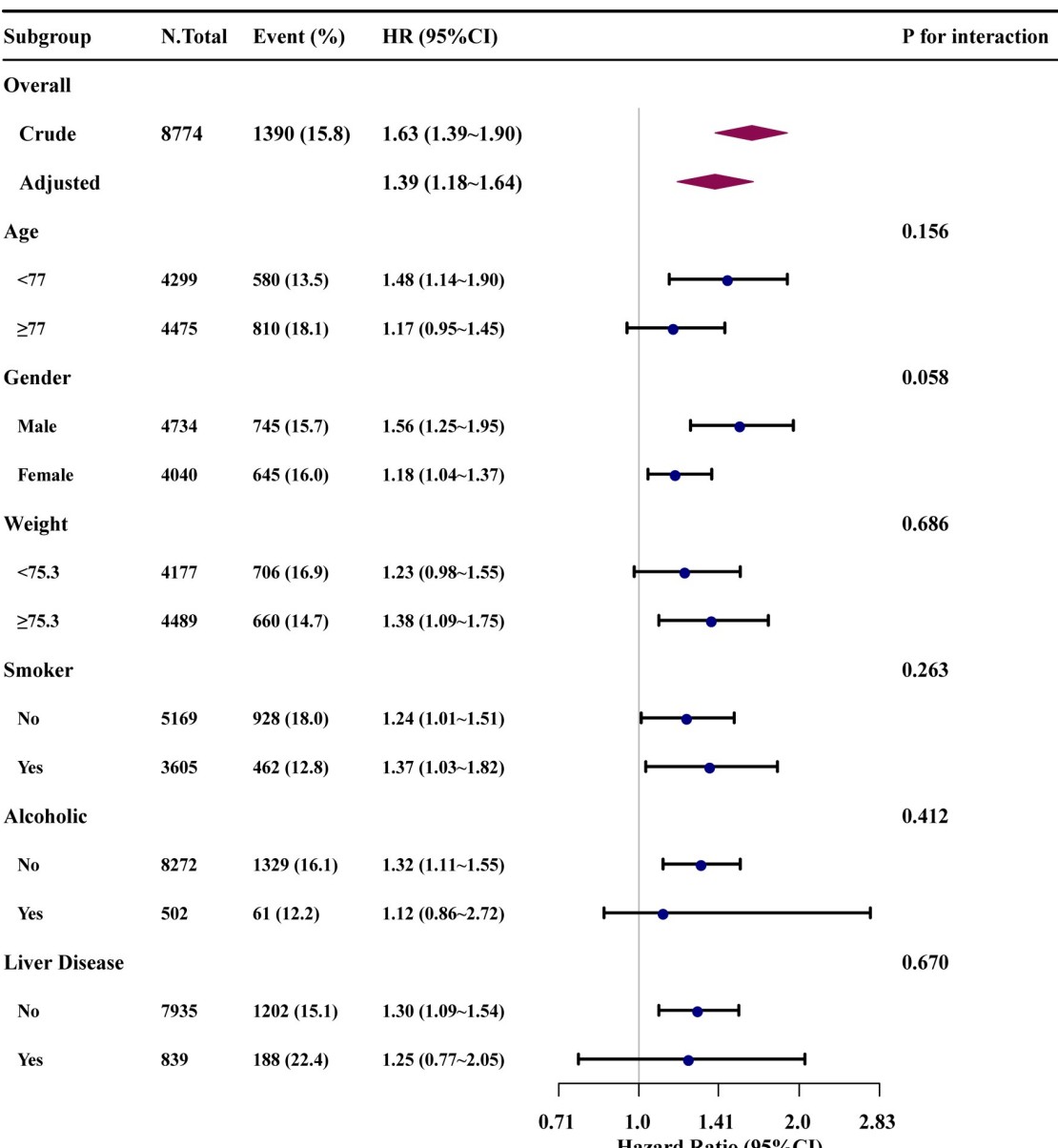

| Subgroup | N.Total | Event (%) | HR (95%CI) | | P for interaction |
|---|---|---|---|---|---|
| **Overall** | | | | | |
| Crude | 8774 | 1390 (15.8) | 1.63 (1.39~1.90) | | |
| Adjusted | | | 1.39 (1.18~1.64) | | |
| **Age** | | | | | 0.156 |
| <77 | 4299 | 580 (13.5) | 1.48 (1.14~1.90) | | |
| ≥77 | 4475 | 810 (18.1) | 1.17 (0.95~1.45) | | |
| **Gender** | | | | | 0.058 |
| Male | 4734 | 745 (15.7) | 1.56 (1.25~1.95) | | |
| Female | 4040 | 645 (16.0) | 1.18 (1.04~1.37) | | |
| **Weight** | | | | | 0.686 |
| <75.3 | 4177 | 706 (16.9) | 1.23 (0.98~1.55) | | |
| ≥75.3 | 4489 | 660 (14.7) | 1.38 (1.09~1.75) | | |
| **Smoker** | | | | | 0.263 |
| No | 5169 | 928 (18.0) | 1.24 (1.01~1.51) | | |
| Yes | 3605 | 462 (12.8) | 1.37 (1.03~1.82) | | |
| **Alcoholic** | | | | | 0.412 |
| No | 8272 | 1329 (16.1) | 1.32 (1.11~1.55) | | |
| Yes | 502 | 61 (12.2) | 1.12 (0.86~2.72) | | |
| **Liver Disease** | | | | | 0.670 |
| No | 7935 | 1202 (15.1) | 1.30 (1.09~1.54) | | |
| Yes | 839 | 188 (22.4) | 1.25 (0.77~2.05) | | |

Hazard Ratio (95%CI)
0.71    1.0    1.41    2.0    2.83

**Fig 4. Subgroup analysis on association of the ASL/ALT ratio with the risk of mortality in patients with an AST/ALT ratio less than 1.80.** Except for the stratification component, each stratification was adjusted for all factors in Model 3.

critically ill older patients among different AST/ALT ratios, suggesting that this ratio may be related to not just liver damage but also systemic disorders.

Additionally, we performed a dose-response analysis, which revealed a non-linear association and saturation effect between the two with an inflection point of 1.80. After adjusting for potential confounding factors, the study found that when the AST/ALT ratio was less than 1.80, the risk of in-hospital mortality increased with an increase in AST/ALT. However, changes in the risk of in-hospital mortality were not statistically significant when the AST/ALT ratio was greater than 1.80. This study is the first, to our knowledge, to discover a non-linear association between the AST/ALT ratio and hospital mortality among critically ill older

patients. Unlike previous research, this study analyzed a dose-response relationship instead of simply demonstrating a linear association between the two.

A previous study found a significant link between lower body weight and higher mortality rates [32]. Furthermore, a higher AST/ALT ratio was linked to alcohol-induced liver damage [33]. Our study was consistent with these findings, as it demonstrated that individuals with a higher AST/ALT ratio tended to have a lower weight and consume more alcohol. Of note, a previous liver disease may affect an individual's AST/ALT ratio substantially. However, by using a multivariable model that adjusted for weight, alcohol, and liver disease, we might explain that none of these confounding factors altered the association between the AST/ALT ratio and mortality.

Although the precise mechanisms underlying the observed relationship between a higher AST/ALT ratio and increased mortality risk are still unknown, there are a few possibilities. First, the AST/ALT ratio may reflect mitochondrial malfunction, which appears to be associated with increased oxidative stress. This theory is supported by research in rats with an elevated AST/ALT ratio, displaying decreased oxygen transport ability and elevated oxidative stress markers [34]. Oxidative stress plays an important role in the development of organ dysfunction and multiple organ failure and leads to multiple organ injuries and high mortality in critically ill patients [35, 36]. Therefore, it stands to reason that an increase in the AST/ALT ratio might be associated with mortality risk by increasing oxidative stress. Second, previous studies have shown a positive association between AST/ALT levels and inflammatory cytokines like CRP, IL-4, IL-6, and TNF-α levels [12, 37]. It suggests that elevated levels of AST/ALT might increase the risk of mortality by exacerbating the inflammatory response. Third, malnutrition, frailty, and sarcopenia are becoming more common in critically ill older patients [38, 39]. Sarcopenia and malnutrition are also major components of the frailty syndrome. Previous research has also revealed an association between higher levels of AST/ALT and worse nutritional indices, implying that AST/ALT may reflect malnutrition, frailty, or sarcopenia [16]. Our study revealed that patients with higher AST/ALT ratios exhibited lower body weights. The inverse relationship between AST/ALT and weight may reflect malnutrition, sarcopenia, and/or frailty. Additional research is necessary to better understand the underlying mechanistic pathways behind the connection between the AST/ALT ratio and the risk of mortality.

Our study is clinically valuable for several reasons. Firstly, the sample size was large enough to adjust for confounding factors and perform subgroup analyses. Secondly, a statistically significant association between AST/ALT and hospital mortality in critically ill older patients was observed only when AST/ALT was less than 1.80. These findings suggest that elderly patients with an elevated AST/ALT ratio may be at higher risk of mortality and require early intervention. Our findings also highlight the significance of monitoring the AST and ALT in elderly patients. Looking at the AST/ALT ratio throughout the hospitalization, in addition to the evaluation of other objective clinical criteria, could aid the clinician in making a more comprehensive, personalized, and informed decision to improve clinical outcomes. Measures such as nutritional intervention, closer monitoring, exercise rehabilitation, aggressive pharmacotherapeutic management, and advanced medical care can be taken to improve their prognosis. However, additional randomized controlled trials are needed to validate this hypothesis.

This study has some limitations: First, our results were obtained in a homogeneous population of critically ill older patients and cannot be extrapolated to other populations. Second, the study design only allowed us to establish statistical associations rather than causation; further prospective studies are required to explore the cause-effect relationships. Third, residual confounding can introduce bias in observational epidemiology analysis, despite performing multivariable analyses. Finally, since elevated AST/ALT levels are associated with poor outcomes, it

is uncertain whether interventions targeted at altering the AST/ALT levels would lead to improved outcomes. Therefore, further prospective and mechanistic studies are needed to validate the findings.

## Conclusions

Taken together, our study demonstrated that the association between the AST/ALT ratio and in-hospital mortality in critically ill older patients was non-linear and exhibited a saturation effect, with an inflection point of 1.80. An elevated AST/ALT ratio was independently associated with increased in-hospital mortality when the AST/ALT ratio was less than 1.80.

## Supporting information

**S1 Table. Association between AST/ALT ratio and the risk of all-cause mortality in different models using imputed data.**
(DOCX)

**S2 Table. Threshold effect analysis of AST/ALT ratio on mortality using two-piecewise linear regression with normal liver function.**
(DOCX)

**S3 Table. Threshold effect analysis of AST/ALT ratio on mortality using two-piecewise linear regression with abnormal liver function.**
(DOCX)

**S1 Fig. The non-linear association between the ASL/ALT ratio and risk of mortality with normal liver function.** Adjusted for all factors in Model 3. The red lines represent the estimated values, and green lines represent their corresponding 95% confidence intervals.
(TIF)

**S2 Fig. The non-linear association between the ASL/ALT ratio and risk of mortality with abnormal liver function.** Adjusted for all factors in Model 3. The red lines represent the estimated values, and green lines represent their corresponding 95% confidence intervals.
(TIF)

**S1 Checklist. STROBE statement—checklist of items that should be included in reports of observational studies.**
(DOCX)

## Acknowledgments

We would like to thank the administrators of the MIMIC-IV database for data support.

## Author Contributions

**Conceptualization:** Bingyan Guo.

**Data curation:** Hua Yang, Limin Meng.

**Formal analysis:** Hua Yang, Limin Meng, Chao Chang.

**Investigation:** Hua Yang, Limin Meng.

**Methodology:** Xiufeng Zhao, Bingyan Guo.

**Writing – original draft:** Hua Yang, Limin Meng.

**Writing – review & editing:** Shuanli Xin, Bingyan Guo.

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
