## [Decision Letter · Decision Letter 0]

10 Aug 2023

PONE-D-23-14780Non-linear association between aspartate aminotransferase to alanine aminotransferase ratio and mortality in critically ill older patients: a retrospective cohort studyPLOS ONE

Dear Dr. Guo,

Thank you for submitting your manuscript to PLOS ONE. After careful consideration, we feel that it has merit but does not fully meet PLOS ONE’s publication criteria as it currently stands. Therefore, we invite you to submit a revised version of the manuscript that addresses the points raised during the review process.

We look forward to receiving your revised manuscript.

Kind regards,

Ahmet Murt

Academic Editor

PLOS ONE

Journal Requirements:

Additional Editor Comments :

Dear authors, we invite you to revise your manuscript given the major revision requests of our independent reviewers.

Reviewers' comments:

Reviewer's Responses to Questions

**Comments to the Author**

1. Is the manuscript technically sound, and do the data support the conclusions?

Reviewer #1: Yes

Reviewer #2: Yes

2. Has the statistical analysis been performed appropriately and rigorously? 

Reviewer #1: Yes

Reviewer #2: Yes

3. Have the authors made all data underlying the findings in their manuscript fully available?

Reviewer #1: Yes

Reviewer #2: Yes

4. Is the manuscript presented in an intelligible fashion and written in standard English?

Reviewer #1: Yes

Reviewer #2: Yes

5. Review Comments to the Author

Reviewer #1: It is an important study in terms of giving real world data on many patients. Especially the adjusted model 3 regression results are interesting. Apart from the ratio, absolute enzyme values can also be an important point. Although there is no need for additional evaluation in the result as it may exceed the purpose of this study, this situation can be mentioned in the discussion. In addition, I think that COVID, which is a disease associated with multiorgan involvement and ICU mortality in the recent period, should have a place in the discussion. This study (DOI: 10.4103/singaporemedj.SMJ-2021-111) , which presents similar data, can be cited.

Reviewer #2: In this manuscript authors studied the relation between AST/ALT ratio and mortality in a cohort of elderly ICU patients. Sample size is large enough for validity of the study. The language and general structure of the manuscript is fine.

However there are some theoretical pitfalls of the article that needs clarification:

1- AST and ALT are enzymes that have certain turnover in cells. They are generally expected to increase with tissue injury, specifically for liver. However median ALT and AST level in this study are in normal range. How can their ratio effect mortality when they are in normal range? I invite the authors to make a separate analysis on patients whose AST and ALT levels are increased.

2- Although their cytoplasmic distribution differ these enzymes tend to deviate from normal in the same direction. However in this study for patients whose AST/ALT ratios increase, we see an increasing pattern in median AST levels and a decreasing pattern in median ALT (Table 1) levels. This should be discussed.

3- Co-morbidities like AKI, sepsis, MI etc are more often in patients who had higher AST/ALT levels. How can mortality be attributed to AST/ALT ratios when there are many other risk factors that increase mortality.

4- What is the utility of this information?

6. PLOS authors have the option to publish the peer review history of their article (what does this mean?). If published, this will include your full peer review and any attached files.

Reviewer #1: No

Reviewer #2: No

---

## [Author Response · Author response to Decision Letter 0]

12 Sep 2023

Below are our specific responses to the editor’s comments.

Responses to the Editor:

1. Please ensure that your manuscript meets PLOS ONE's style requirements, including those for file naming. The PLOS ONE style templates can be found at https://journals.plos.org/plosone/s/file?id=wjVg/PLOSOne_formatting_sample_main_body.pdf and https://journals.plos.org/plosone/s/file?id=ba62/PLOSOne_formatting_sample_title_authors_affiliations.pdf. 

Response 1: We followed the editor’s suggestions and made the corresponding changes to the revised version. 

2. Additional Editor Comments: Dear authors, we invite you to revise your manuscript given the major revision requests of our independent reviewers.

 Response 2: We thank you very much for giving us the opportunity to revise the manuscript, and we have now revised it in accordance with the reviewer’s request.

Below are our specific responses to the reviewers’ comments.

Responses to Reviewer #1: 

Comment 1: 

It is an important study in terms of giving real world data on many patients. Especially the adjusted model 3 regression results are interesting. Apart from the ratio, absolute enzyme values can also be an important point. Although there is no need for additional evaluation in the result as it may exceed the purpose of this study, this situation can be mentioned in the discussion. In addition, I think that COVID, which is a disease associated with multiorgan involvement and ICU mortality in the recent period, should have a place in the discussion. This study (DOI: 10.4103/singaporemedj.SMJ-2021-111), which presents similar data, can be cited.

Response 1: 

Thank you very much for your careful review and constructive suggestions with regard to our manuscript. 

First, I totally agree with your point of view that, apart from the ratio, absolute enzyme values can also be an important point. We have added the content to the discussion section of the revised manuscript. Here are the details: According to recent research, ALT is a more specific measure for the liver compared to AST. AST tends to increase or remain stable with age, whereas ALT generally decreases in elderly populations [27, 28]. Several studies have reported that low ALT levels in older individuals can be considered a marker of age-related frailty, disability, malnutrition, and sarcopenia, and are associated with higher all-cause mortality [14, 29]. This finding could be explained by age-related decline, which subsequently increases susceptibility to diseases. By contrast, AST is known to be an enzyme released from different tissues, reflecting anaerobic glycolysis. People with increased AST levels may have more severe tissue damage. A recent meta-analysis study with aggregate data found a comparatively moderate association between AST and all-cause mortality in general populations [30]. However, another study showed AST was positively associated with long-term mortality in this elderly population [31]. As a result, decreased ALT and increased AST levels could result in a high AST/ALT ratio. As observed in this study, there is an increasing pattern in median AST levels and a decreasing pattern in median ALT levels for patients whose AST/ALT ratios increase. (Page 15, lines 269-277, and Page 16, lines 278-283)

Second, according to the reviewer’s comment, we have added additional information about COVID in the “Discussion” section as follows: COVID, which is a disease associated with multiorgan involvement and ICU mortality in recent years. According to the former study, it showed that increasing AST and an initial AST to ALT ratio >1.5 are associated with mortality in hospitalized COVID-19 patients and may be an early warning [23]. (Page 16, lines 296-298, and Page 17, lines 299-300) 

References

23. Bakkaloglu OK, Onal U, Eskazan T, Kurt EA, Candan S, Karaali R, et al. Increase in transaminase levels during COVID-19 infection and its association with poor prognosis. Singapore Med J. 2023. https://doi.org/10.4103/singaporemedj.SMJ-2021-111 PMID: 37171429.

27. Preuss HG, Kaats GR, Mrvichin N, Bagchi D, Preuss JM. Circulating ALT Levels in Healthy Volunteers Over Life-Span: Assessing Aging Paradox and Nutritional Implications. J Am Coll Nutr. 2019;38(8):661-9. https://doi.org/10.1080/07315724.2019.1580169 PMID: 31075051.

28. Ke P, Zhong L, Peng W, Xu M, Feng J, Tian Q, et al. Association of the serum transaminase with mortality among the US elderly population. J Gastroenterol Hepatol. 2022;37(5):946-53. https://doi.org/10.1111/jgh.15815 PMID: 35233823.

29. Le Couteur DG, Blyth FM, Creasey HM, Handelsman DJ, Naganathan V, Sambrook PN, et al. The association of alanine transaminase with aging, frailty, and mortality. J Gerontol A Biol Sci Med Sci. 2010;65(7):712-7. https://doi.org/10.1093/gerona/glq082 PMID: 20498223.

30. Kunutsor SK, Apekey TA, Seddoh D, Walley J. Liver enzymes and risk of all-cause mortality in general populations: a systematic review and meta-analysis. Int J Epidemiol. 2014;43(1):187-201. https://doi.org/10.1093/ije/dyt192 PMID: 24585856.

31. Koehler EM, Sanna D, Hansen BE, van Rooij FJ, Heeringa J, Hofman A, et al. Serum liver enzymes are associated with all-cause mortality in an elderly population. Liver Int. 2014;34(2):296-304. https://doi.org/10.1111/liv.12311 PMID: 24219360.

Responses to Reviewer #2: 

In this manuscript authors studied the relation between AST/ALT ratio and mortality in a cohort of elderly ICU patients. Sample size is large enough for validity of the study. The language and general structure of the manuscript is fine.

However there are some theoretical pitfalls of the article that needs clarification:

 Comment 1: 

1- AST and ALT are enzymes that have certain turnover in cells. They are generally expected to increase with tissue injury, specifically for liver. However median ALT and AST level in this study are in normal range. How can their ratio effect mortality when they are in normal range? I invite the authors to make a separate analysis on patients whose AST and ALT levels are increased.

 Response 1: 

Thank you very much for your careful review and constructive suggestions with regard to our manuscript. Meanwhile, I am also very grateful to you for the specific modification strategy. We have made a separate analysis on patients whose AST and ALT levels are increased according to your suggestion. We have added the content to the result section of the revised manuscript. The details are as follows: 

In addition, we divided the total population into two groups: one with normal values of liver function indicators (AST ≤ 40 U/L and ALT ≤ 40 U/L) and the other with abnormal liver function indicators (AST > 40 U/L or ALT > 40 U/L). We have evaluated the non-linear relationship and used a two-piecewise linear regression model to determine the inflection point. Similarly, a non-linear relationship and saturation effect were observed between them, respectively. The results were listed in S2 and S3 Tables, S1 and S2 Figs. (Page 13, lines 222-228)

S1 Fig. The non-linear association between the ASL/ALT ratio and risk of mortality with normal liver function. Adjusted for all factors in Model 3. The red lines represent the estimated values, and green lines represent their corresponding 95% confidence intervals. (Page 21, lines 384-387)

S2 Fig. The non-linear association between the ASL/ALT ratio and risk of mortality with abnormal liver function. Adjusted for all factors in Model 3. The red lines represent the estimated values, and green lines represent their corresponding 95% confidence intervals. (Page 21, lines 388-391)

S2 Table Threshold effect analysis of AST/ALT ratio on mortality using two-piecewise linear regression with normal liver function. (Page 20, lines 380-381)

 HR (95% CI) P value

Fitting model by standard linear regression 1.07 (0.97∼1.17) 0.187

Fitting model by two-piece wise linear regression 

Inflection point of the AST/ALT ratio 1.80 

 ≤ 1.80 1.21 (1.10∼1.50) < 0.001

> 1.80 1.01 (0.83∼1.15) 0.780

P for log likelihood ratio test < 0.001

Notes: We adjusted for: age, gender, smoking, alcoholic, weight, systolic blood pressure, respiratory rate, liver disease, cerebrovascular disease, cardiac arrest, cardiogenic shock, acute kidney injury, sequential organ failure assessment, systemic inflammatory response syndrome, hemoglobin, platelets, anion gap, blood urea nitrogen, and potassium.

Abbreviations: HR, hazard ratios; CI, confidence intervals.

S3 Table Threshold effect analysis of AST/ALT ratio on mortality using two-piecewise linear regression with abnormal liver function. (Page 21, lines 382-383)

 HR (95% CI) P value

Fitting model by standard linear regression 1.04 (1.01∼1.08) 0.047

Fitting model by two-piece wise linear regression 

Inflection point of the AST/ALT ratio 1.80 

 ≤ 1.80 1.51 (1.15∼1.99) < 0.001

> 1.80 1.01 (0.96∼1.06) 0.694

P for log likelihood ratio test < 0.001

Notes: We adjusted for: age, gender, smoking, alcoholic, weight, systolic blood pressure, respiratory rate, liver disease, cerebrovascular disease, cardiac arrest, cardiogenic shock, acute kidney injury, sequential organ failure assessment, systemic inflammatory response syndrome, hemoglobin, platelets, anion gap, blood urea nitrogen, and potassium.

Abbreviations: HR, hazard ratios; CI, confidence intervals.

Comment 2: 

2- Although their cytoplasmic distribution differ these enzymes tend to deviate from normal in the same direction. However, in this study for patients whose AST/ALT ratios increase, we see an increasing pattern in median AST levels and a decreasing pattern in median ALT (Table 1) levels. This should be discussed.

 Response 2: 

We thank the reviewer for pointing out this issue. In our study, elderly patients were selected as subjects, and given that AST increases or stabilizes with age while ALT decreases with age in general elderly populations, decreased ALT and increased AST levels might be responsible for the increased AST/ALT ratio. We have added the content to the discussion section of the revised manuscript. The details are as follows: According to recent research, ALT is a more specific measure for the liver compared to AST. AST tends to increase or remain stable with age, whereas ALT generally decreases in elderly populations [27, 28]. Several studies have reported that low ALT levels in older individuals can be considered a marker of age-related frailty, disability, malnutrition, and sarcopenia, and are associated with higher all-cause mortality [14, 29]. This finding could be explained by age-related decline, which subsequently increases susceptibility to diseases. By contrast, AST is known to be an enzyme released from different tissues, reflecting anaerobic glycolysis. People with increased AST levels may have more severe tissue damage. A recent meta-analysis study with aggregate data found a comparatively moderate association between AST and all-cause mortality in general populations [30]. However, another study showed AST was positively associated with long-term mortality in this elderly population [31]. As a result, decreased ALT and increased AST levels could result in a high AST/ALT ratio. As observed in this study, there is an increasing pattern in median AST levels and a decreasing pattern in median ALT levels for patients whose AST/ALT ratios increase. (Page 15, lines 269-277, and Page 16, lines 278-283)

Comment 3: 

3- Co-morbidities like AKI, sepsis, MI etc are more often in patients who had higher AST/ALT levels. How can mortality be attributed to AST/ALT ratios when there are many other risk factors that increase mortality.

 Response 3: 

We thank the reviewer for pointing out this issue. Firstly, we sincerely apologize for any confusion caused by our inappropriate wording in the original manuscript. According to the reviewer, we have replaced “contribute to” with “associated with” in the revised manuscript. The details are as follows: Therefore, it stands to reason that an increase in the AST/ALT ratio might be associated with mortality risk by increasing oxidative stress. (Page 18, lines 332) 

In addition, we also replaced “reflect” with “related” in the revised manuscript. The details are as follows: In agreement with these findings, we found a significant difference in the in-hospital mortality of critically ill older patients among different AST/ALT ratios, suggesting that this ratio may be related to not just liver damage but also systemic disorders. (Page 17, lines 303)

Secondly, co-morbidities like AKI, sepsis, MI, etc. are more common in patients with higher AST/ALT levels. However, this study explored association but not causation due to the design of a retrospective cohort study. The data shown are only correlations, not cause-and-effect relationships. Clearly, independent association does not imply causality, and there are no easily discernible mechanisms by which an increased AST/ALT ratio would contribute to the increased mortality risk. Based on your suggestion, we have added it to the limitations as “Second, the study design only allowed us to establish statistical associations rather than causation; further prospective studies are required to explore the cause-effect relationships.” (Page 20, lines 363-365)

Comment 4: 

4- What is the utility of this information?

Response 4: 

We would like to thank the reviewer for your thoughtful review of our manuscript. These findings suggest that elderly patients with an elevated AST/ALT ratio may be at higher risk of mortality and require early intervention. Therefore, when patients are on ICU admission and under treatment, clinicians should pay more attention to the AST/ALT ratio in critically ill elderly patients. Measures such as nutritional intervention, closer monitoring, aggressive pharmacotherapeutic management, and advanced medical care can be taken to improve their clinical outcomes. We fully agree with your suggestion and have added this part to the “Discussion” part as follows: Our findings highlight the significance of monitoring the AST and ALT in elderly patients. Looking at the AST/ALT ratio throughout the hospitalization, in addition to the evaluation of other objective clinical criteria, could aid the clinician in making a more comprehensive, personalized, and informed decision to improve clinical outcomes. Measures such as nutritional intervention, closer monitoring, exercise rehabilitation, aggressive pharmacotherapeutic management, and advanced medical care can be taken to improve their prognosis. (Page 19, lines 351-358)

We appreciate your efforts in reviewing our manuscript, which have made our study clearer and more comprehensive. We believe that we have adequately responded to the editor’s and reviewers’ comments, and we hope that our paper is now acceptable for publication by PLOS ONE.

---

## [Decision Letter · Decision Letter 1]

19 Oct 2023

Non-linear association between aspartate aminotransferase to alanine aminotransferase ratio and mortality in critically ill older patients: a retrospective cohort study

PONE-D-23-14780R1

Dear Dr. Guo,

We’re pleased to inform you that your manuscript has been judged scientifically suitable for publication and will be formally accepted for publication once it meets all outstanding technical requirements.

Kind regards,

Ahmet Murt

Academic Editor

PLOS ONE

Additional Editor Comments (optional):

None.

Reviewers' comments:

Reviewer's Responses to Questions

**Comments to the Author**

1. If the authors have adequately addressed your comments raised in a previous round of review and you feel that this manuscript is now acceptable for publication, you may indicate that here to bypass the “Comments to the Author” section, enter your conflict of interest statement in the “Confidential to Editor” section, and submit your "Accept" recommendation.

Reviewer #1: All comments have been addressed

Reviewer #2: All comments have been addressed

2. Is the manuscript technically sound, and do the data support the conclusions?

Reviewer #1: Yes

Reviewer #2: Yes

3. Has the statistical analysis been performed appropriately and rigorously? 

Reviewer #1: Yes

Reviewer #2: Yes

4. Have the authors made all data underlying the findings in their manuscript fully available?

Reviewer #1: Yes

Reviewer #2: Yes

5. Is the manuscript presented in an intelligible fashion and written in standard English?

Reviewer #1: Yes

Reviewer #2: Yes

6. Review Comments to the Author

Reviewer #1: I thank the authors for their strenous effort to answer all the suggestions of reviewers. All concerns are adequately addressed.

Reviewer #2: In this revised version of the manuscript, all of my concerns are well answered. I have no further comments.

7. PLOS authors have the option to publish the peer review history of their article (what does this mean?). If published, this will include your full peer review and any attached files.

Reviewer #1: No

Reviewer #2: No

---

## [Editor Report · Acceptance letter]

24 Oct 2023

PONE-D-23-14780R1 

Non-linear association between aspartate aminotransferase to alanine aminotransferase ratio and mortality in critically ill older patients: a retrospective cohort study 

Dear Dr. Guo:

I'm pleased to inform you that your manuscript has been deemed suitable for publication in PLOS ONE. Congratulations! Your manuscript is now with our production department. 

Kind regards, 

on behalf of

Dr. Ahmet Murt 

Academic Editor

PLOS ONE